# Rat and mouse cardiomyocytes show subtle differences in creatine kinase expression and compartmentalization

**Jelena Branovets, Kärol Soodla, Marko Vendelin, Rikke Birkedal***

Laboratory of Systems Biology, Department of Cybernetics, Tallinn University of Technology, Tallinn, Estonia

\* rikke@sysbio.ioc.ee

## Abstract

Creatine kinase (CK) and adenylate kinase (AK) are energy transfer systems. Different studies on permeabilized cardiomyocytes suggest that ADP-channelling from mitochondrial CK alone stimulates respiration to its maximum, $V_{O2\_max}$, in rat but not mouse cardiomyocytes. Results are ambiguous on ADP-channelling from AK to mitochondria. This study was undertaken to directly compare the CK and AK systems in rat and mouse hearts. In homogenates, we assessed CK- and AK-activities, and the CK isoform distribution. In permeabilized cardiomyocytes, we assessed mitochondrial respiration stimulated by ADP from CK and AK, $V_{O2\_CK}$ and $V_{O2\_AK}$, respectively. The ADP-channelling from CK or AK to mitochondria was assessed by adding PEP and PK to competitively inhibit the respiration rate. We found that rat compared to mouse hearts had a lower aerobic capacity, higher $V_{O2\_CK}/V_{O2\_max}$, and different CK-isoform distribution. Although rat hearts had a larger fraction of mitochondrial CK, less ADP was channeled from CK to the mitochondria. This suggests different intracellular compartmentalization in rat and mouse cardiomyocytes. $V_{O2\_AK}/V_{O2\_max}$ was similar in mouse and rat cardiomyocytes, and AK did not channel ADP to the mitochondria. In the absence of intracellular compartmentalization, the AK- and CK-activities in homogenate should have been similar to the ADP-phosphorylation rates estimated from $V_{O2\_AK}$ and $V_{O2\_CK}$ in permeabilized cardiomyocytes. Instead, we found that the ADP-phosphorylation rates estimated from permeabilized cardiomyocytes were 2 and 9 times lower than the activities recorded in homogenate for CK and AK, respectively. Our results highlight the importance of energetic compartmentalization in cardiac metabolic regulation and signalling.

## Introduction

Creatine kinase (CK) is thought to play a crucial role in storage and spatial transport of energy-rich phosphates in tissues with high and fluctuating energy demands [1, 2]. In the heart, there are three cytosolic CK isoforms (MM-, MB-, and BB-CK) and one mitochondrial CK isoform (Mi-CK). Several studies on rat heart have reported that phosphotransfer via CK is several times faster than ATP synthesis by oxidative phosphorylation in mitochondria [3–5], and Mi-CK in the intermembrane space is coupled to the adenine nucleotide translocase (ANT), which exchanges ATP for ADP, providing ADP for mitochondrial oxidative

**Data Availability Statement:** All relevant data are within the paper and its Supporting information files.

**Funding:** This work was supported by the Estonian Research Council (www.etag.ee/en/), grant number PRG1127. The funder had no role in study design, data collection and analysis, decision to publish, or preparation of the manuscript.

**Competing interests:** The authors have declared that no competing interests exist.

**Abbreviations:** AGAT, Arginine:glycine amidinotransferase (EC 2.1.4.1); AK, Adenylate kinase; $AK_{ADP\_GMPS}$, Name of the protocol used to record adenylate kinase-stimulated respiration with glutamate, malate, pyruvate, and succinate as substrates, followed by stimulation of maximal respiration with 2mM ADP; $AK_f$, Adenylate kinase activity measured in the forward direction: ATP + AMP → ADP + ADP; $AK_{PEP/PK\_GMPS}$, Name of the protocol used to record adenylate kinase-stimulated respiration with glutamate, malate, pyruvate, and succinate as substrates, followed by inhibition of respiration with phosphoenolpyruvate and pyruvate kinase; $AK_r$, Adenylate kinase activity measured in the reverse direction: ADP + ADP → ATP + AMP; AMPK, AMP-activated protein kinase; ANT, adenine nucleotide translocase; BB-CK, Homodimeric brain isoform of creatine kinase; CK, Creatine kinase; $CK_{ADP\_GM}$, Name of the protocol used to record creatine kinase-stimulated respiration with glutamate and malate as substrates, followed by stimulation of maximal respiration with 2mM ADP; $CK_{ADP\_GMPS}$, Name of the protocol used to record creatine kinase-stimulated respiration with glutamate, malate, pyruvate, and succinate as substrates, followed by stimulation of maximal respiration with 2mM ADP; $CK_f$, Creatine kinase activity measured in the forward direction:ATP + creatine → ADP + phosphocreatine + $H^+$; $CK_{PEP/PK\_GM}$, Name of the protocol used to record creatine kinase-stimulated respiration with glutamate and malate as substrates, followed by inhibition of respiration with phosphoenolpyruvate and pyruvate kinase; $CK_{PEP/PK\_GMPS}$, Name of the protocol used to record creatine kinase-stimulated respiration with glutamate, malate, pyruvate, and succinate as substrates, followed by inhibition of respiration with phosphoenolpyruvate and pyruvate kinase; $CK_r$, Creatine kinase activity measured in the reverse direction:ADP + phosphocreatine + $H^+$ → ATP + creatine; CM, Cardiomyocytes; CO, Cytochrome oxidase; CS, Citrate synthase; Cyt $aa_3$, Cytochrome $aa_3$; Cyt c, Cytochrome c; DTNB, dithiobis(2-nitrobenzoic acid); FCCP, Carbonyl cyanide 4-trifluoromethoxyphenylhydrazone; GAMT, Guanidinoacetate N-methyltransferase (EC 2.1.1.2); GM, Glutamate and malate; GMPS, Glutamate, malate, pyruvate and succinate; MB-CK,

phosphorylation [6, 7]. For many years, the CK system was considered to have a pivotal role in the regulation of cardiac energy metabolism [1, 2, 8–10].

Surprisingly, studies on genetically manipulated mouse models having modifications in various components of the CK system like expression of different CK isoforms, creatine synthesis or uptake have been equivocal [11]. Notably, the baseline cardiac function of mice with compromised CK system is minimally disturbed [12–16] and compromised CK function does not exacerbate heart failure [17, 18]. Thus, studies on mice have not corroborated the theory, based mainly on rat experiments, that CK is crucial for energy transfer and regulation of oxidative phosphorylation.

Our recent study on permeabilized cardiomyocytes from creatine-deficient AGAT [19] and GAMT [20] mice also pointed to a difference between mouse and rat cardiomyocytes. We performed an assay, where we recorded the respiration rate of permeabilized cardiomyocytes. In addition to substrates for oxidative phosphorylation (glutamate and malate, GM) and creatine, we added ATP to the solution to initiate endogenous ADP-generation by CK (creatine + ATP → phosphocreatine + ADP + $H^+$). Endogenous ADP then stimulated oxidative phosphorylation. We found that the rate of ADP-generation by CK sustained a respiration rate that was ~80% of the maximal respiration rate [21]. The channelling of ADP from CK to the mitochondria was assessed by addition of phosphoenolpyruvate (PEP) and pyruvate kinase (PK) in excess, which compete with the mitochondria for ADP. This is an experimental strategy to assess how much of the CK is so closely associated with the mitochondria that ADP is channelled directly to the mitochondria without being released to the bulk phase, where it would be consumed by PK. The addition of PEP and PK lowered the rate of oxidative phosphorylation by ~75%, demonstrating that some ADP generated by CK is channelled to the mitochondria and inaccessible to PK [21]. However, our results on mouse cardiomyocytes were in sharp contrast to experiments on rat cardiomyocytes, where even in the presence of PEP and PK, the rate of ADP-generation by CK sustained the respiration rate at its maximum [22, 23]. This difference raised the question whether rat and mouse hearts differ in terms of their CK activity, either total and/or relative to the maximal respiration rate in the absence and presence of PEP and PK.

Adenylate kinase (AK) is another well-known alternative phosphotransfer system in the heart [9]. Its role in facilitating energy transfer and in the compartmentalization of adenine nucleotides has also been investigated [24–29]. Although the contribution of AK-mediated phosphoryl transfer to the total ATP turnover is only ~10%, some studies observed an increased importance of AK under stress conditions [29, 30]. In our recent study on mouse cardiomyocytes [21], we also assessed the respiration rate stimulated by endogenous ADP generated by AK. In the presence of substrates (GM) and ATP, the addition of AMP initiated the endogenous ADP-generation by AK (AMP + ATP → 2 ADP). The rate of ADP-generation by AK sustained the respiration rate near its maximal rate, but the addition of PEP and PK abolished the effect of adding AMP, suggesting that all ADP generated by AK was accessible to PK. On the one hand, this is in agreement with the finding of a minimal AK activity in rat and mouse heart mitochondria [31–33]. On the other hand, it contradicts results demonstrating a stronger functional coupling between AK and mitochondrial respiration in rats [27]. Thus, there is an inexplicable mismatch between different studies regarding the importance of mitochondrial AK in regulation of mitochondrial oxidative phosphorylation, and we speculated whether species differences might be adding to the confusion.

The aim of the present study was to assess whether rat and mouse hearts are different in terms of 1) the overall CK and AK activities in whole heart homogenates, and 2) how much endogenous ADP generated by CK or AK stimulates oxidative phosphorylation in the absence and presence of PEP and PK competing with the mitochondria for the consumption of ADP.

Heterodimeric muscle-brain isoform of creatine kinase; Mi-CK, Sarcomeric mitochondrial creatine kinase; MM-CK, Homodimeric muscle isoform of creatine kinase; $P/O_2$ ratio, The phosphate to oxygen ratio, describes how much ADP the mitochondria phosphorylate to ATP per $O_2$ consumed; PEP, Phosphoenolpyruvate; PK, Pyruvate kinase; SERCA, sarcoendoplasmic reticulum $Ca^{2+}$-ATPase; TNB, thionitrobenzoate; $V_{ADP\_AK}$, Estimated rate of ADP-phosphorylation, when mitochondria are stimulated by ADP generated by adenylate kinase; $V_{ADP\_CK}$, Estimated rate of ADP-phosphorylation, when mitochondria are stimulated by ADP generated by creatine kinase; $V_{ADP\_max}$, Estimated rate of ADP-phosphorylation, when mitochondria are stimulated by 2 mM exogenous ADP; VDAC, Voltage dependent anion channel; $V_{FCCP}$, Respiration rate during maximal uncoupled respiration rate; $V_{O2\_AK}$, Respiration rate stimulated by adenylate kinase after addition of ATP and AMP; $V_{O2\_CK}$, Respiration rate stimulated by creatine kinase after addition of ATP with creatine in the medium; $V_{O2\_max}$, Maximal coupled respiration rate stimulated by 2 mM ADP; $V_{O2\_max\_GM}$, Maximal coupled respiration rate with glutamate and malate as substrates; $V_{O2\_max\_GMPS}$, Maximal coupled respiration rate with glutamate, malate, pyruvate and succinate as substrates.

**Table 1. Characteristics of the mice and rats used in the experiments.**

|  | n | BW, g | HW, mg | HW/BW, mg/g |
|---|---|---|---|---|
| **Mice** | 15 (7) | 27.3 ± 0.8 | 123.8 ± 2.3 | 4.7 ± 0.1 |
| **Rats** | 14 (7) | 316.6 ± 8 | 920.4 ± 26 | 2.9 ± 0.1 |

Values are shown as mean ± SEM. BW, body weight; HW, absolute heart weight; HW/BW, relative heart weight. The total number of animals is shown in column n. The absolute and relative heart weight (HW and HW/BW) are reported for a smaller number of animals, indicated in parenthesis, because the hearts used for cardiomyocyte isolation could not be weighed.

In whole heart homogenates, we recorded the activities of CK and AK in the direction of both ADP- and ATP-generation. In addition, we recorded the activities of citrate synthase (CS, a marker of mitochondrial density) and cytochrome oxidase (CO, a marker of oxidative capacity, i.e. maximal $O_2$ consumption rate per muscle mass) [34]. In permeabilized cardiomyocytes, we recorded the respiration rate stimulated by endogenous ADP generated by either CK or AK. In one set of experiments, we assessed how much CK or AK stimulated respiration relative to its maximal rate by subsequent addition of ADP. In parallel, the ADP channelling between kinases and mitochondria was assessed by subsequent additions of PEP and PK.

The maximal respiration rate is substrate-dependent [35, 36]. Most studies have used only GM, which lead to NADH-linked electron flux only through complex I of the respiratory chain. More substrates are required to obtain the maximal respiration rate. For example, a combination of glutamate, malate, pyruvate and succinate (GMPS) leads to NADH and FAD-linked electron flux through both complexes I and II of the respiratory chain [36, 37]. Here, the role of CK was assessed with GM as in previous experiments [21–23], and with GMPS. Due to a limited number of chambers in the respirometer, the role of AK was only assessed with GMPS. To the best of our knowledge, this is the first time these recordings have been performed with GMPS.

Throughout this work, we compare whole heart homogenate with isolated cardiomyocytes. However, whole heart homogenates are prepared using pieces of tissue, which in addition to cardiomyocytes also contain extracellular matrix, endothelial cells, and fibroblasts. Cardiomyocytes, endothelial cells and fibroblasts take up 70–80%, 3.2–5.3%, and 1.4–1.9% of the volume, respectively [38], and cardiomyocytes have a much larger mitochondrial volume (31–40%) than the other cell types (5% for endothelial cells) [39–41]. Therefore, in order to compare whole heart homogenate and isolated cardiomyocytes, we assumed that the CS activity in non-cardiomyocytes was negligible and compared the data normalized to the CS activity.

## Results

### Animals and cell preparations

The morphological characteristics of the animals used in this study are given in Table 1. Table 2 shows several characteristics of the cell suspensions. The viability of the cell

**Table 2. Characteristics of the isolated cardiomyocyte suspensions from mice and rats.**

|  | Viability % | Protein mg/ml | CS activity µmol/min/g protein | Cyt aa₃ µmol/g protein |
|---|---|---|---|---|
| Mouse cardiomyocytes | 75.0 ± 1.1 | 15.52 ± 0.80 | 806 ± 37 |  |
| Rat cardiomyocytes | 73.7 ± 1.9 | 20.80 ± 2.41 | 609 ± 9 *** | 0.16 ± 0.01 |

Cardiomyocytes were isolated from 8 mice and 7 rats. Values are shown as mean ± SEM. Viability is the fraction of rod-shaped to the total number of cells. CS is the citrate synthase activity, and cyt $aa_3$ is the cytochrome $aa_3$ content.

*** denotes $p < 0.001$, significant difference between species.

suspensions did not differ between mice and rats. Although there was a tendency to a higher protein content in rats than in mice, this was not statistically significant. CS activity, when normalized to the protein content, was significantly higher in mouse than in rat cardiomyocytes. The cytochrome $aa_3$ (Cyt $aa_3$) content was assessed only for cell suspensions from rat hearts due to the large volume required for these measurements.

### Enzyme activities and CK isoform distribution

The overall activities of CS, CK, AK and CO, measured in cardiac whole tissue homogenates, are shown in Fig 1. In Fig 1A, the enzymatic activities normalized to the heart wet weight are shown for comparison with the literature (see Discussion). In Fig 1B, the activities of CO, AK and CK normalized to the CS activity are shown for comparison with the respiration rates recorded in permeabilized cardiomyocytes (Figs 3 and 4). Since both CK- and AK-catalysed phosphotransfer reactions are reversible, the activities of these enzymes were

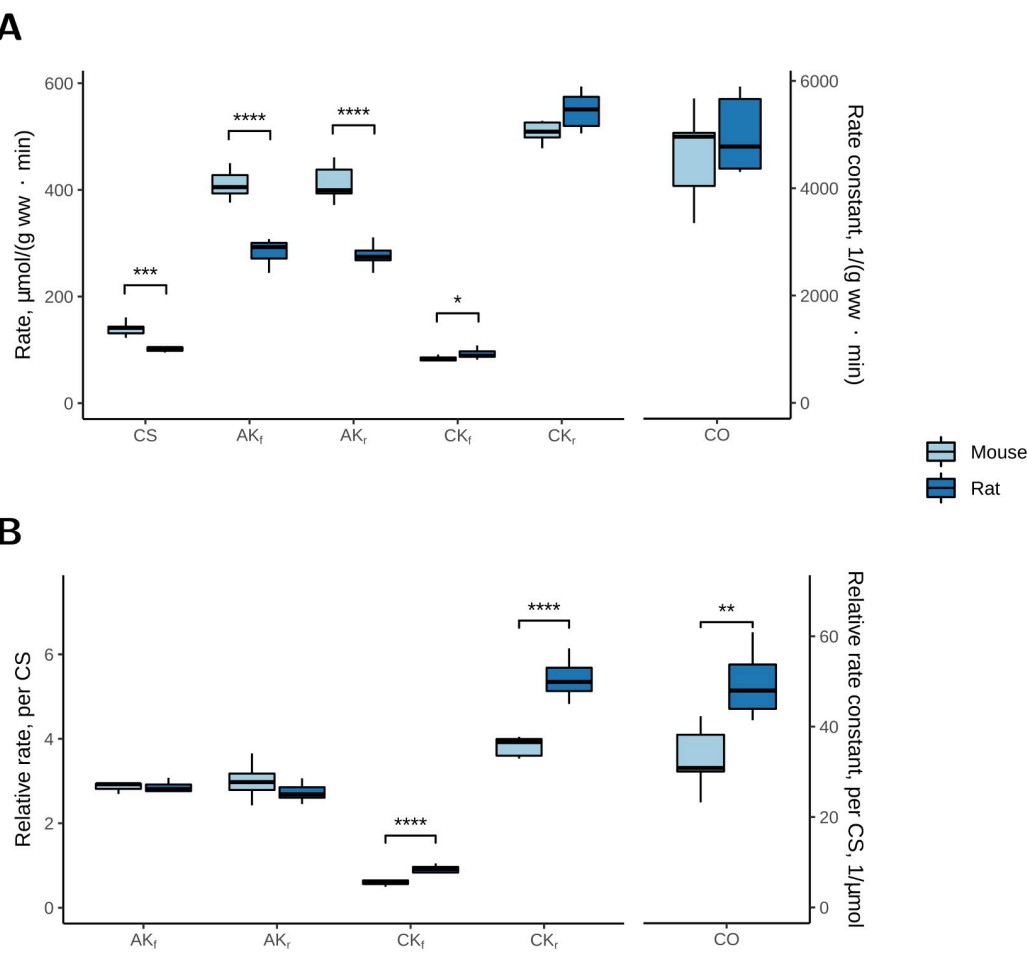

**Fig 1. The activities of citrate synthase (CS), adenylate kinase (AK), creatine kinase (CK) and cytochrome oxidase (CO) in mouse and rat hearts.** CS, AK and CK activities are represented as rates ($\mu mol \cdot min^{-1} \cdot g\ ww^{-1}$) and CO activity as a rate constant ($min^{-1} \cdot g\ ww^{-1}$). $AK_f$ and $CK_f$, and $AK_r$ and $CK_r$, are the forward (f) and reverse (r) reaction rates of AK and CK, respectively. *A*: When normalized to the wet weight of the tissue, CS and AK activities were higher in mouse than in rat heart. *B*: When normalized to the CS activity, the CK and CO activities ($\mu mol \cdot min^{-1} \cdot IU\ CS^{-1}$) were higher in rat than in mouse heart. The number of animals was $n = 7$ for mice and $n = 7$ for rats. * denotes $p < 0.05$, ** $p < 0.01$, *** $p < 0.001$, **** $p < 0.0001$ significant difference between species.

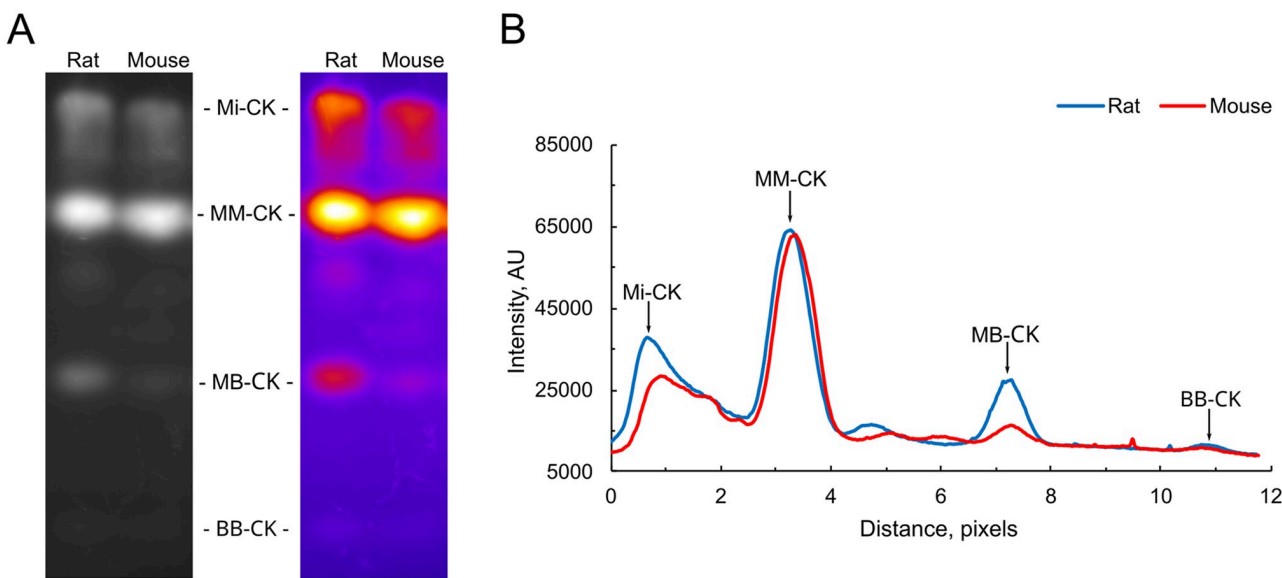

**Fig 2. The CK isoform distribution in rat and mouse hearts.** The CK isoform distribution in rat and mouse hearts was assessed by agarose gel electrophoresis. A: Representative picture showing on the left the raw image, and, on the right the same image, in pseudocolour to highlight the bands. B. The intensity profiles from the rat and mouse lanes shown in A.

measured in both directions. In the forward direction, CK and AK generate ADP, i.e. CK catalyses the reaction: creatine + ATP → phosphocreatine + H$^+$ + ADP, and AK catalyses the reaction: ATP + AMP → 2 ADP. In the reverse direction, CK catalyses the reaction: phosphocreatine + H$^+$ + ADP → creatine + ATP, and AK catalyses the reaction: 2 ADP → ATP + AMP.

When enzyme activities were normalized to the wet weight (Fig 1A), the CS activity was ~35% higher in mice than in rats (139.1 ± 4.7 vs 101.4 ± 1.6 µmol·min-1·g ww-1 respectively; $p < 0.001$). The AK activity was ~30% higher in mice than in rats in both directions ($p < 0.001$). The CK$_r$ activity in the direction of ATP-production tended to be higher in rat than in mouse heart ($p = 0.0815$), and the CK$_f$ activity measured in the direction of ADP-production was 11% lower in mice than in rats ($p < 0.05$). The CO activity was similar for mice and rats.

As the CS activity was different in mouse and rat hearts, normalizing hereto changed the pattern (Fig 1B). The AK/CS activity (measured in both directions) was similar in mice and rats, whereas the CK/CS activity was ~35% lower in mice than in rats in both directions (Fig 1B; $p < 0.0001$), and CO/CS activity was ~30% lower in mice than in rats (Fig 1B; $p < 0.01$).

The CK isoform distribution in the hearts of mice and rats was assessed by gel electrophoresis. A representative picture is shown in Fig 2. For each lane, the intensity of the four major bands corresponding to Mi-, MM-, MB and BB-CK was quantified, and the fractional intensity of each band was calculated. The averaged data are given in Table 3. Compared to mouse hearts, rat hearts had a different CK isoform distribution with a smaller fraction of MM-CK, and larger fractions of Mi-CK, MB-CK and BB-CK.

## Stimulation of respiration by CK and AK

In permeabilized cardiomyocytes, the stimulation of respiration by CK or AK was assessed relative to the maximal coupled respiration rate (V$_{O2}$_max) (Fig 3). CK-stimulated

**Table 3. Distribution of CK isoforms in the hearts of mice and rats.**

| | Mi-CK, % | MM-CK, % | MB-CK, % | BB-CK, % |
|---|---|---|---|---|
| **Mouse heart** | 31.6 ± 1.8 | 63.8 ± 1.6 | 3.7 ± 0.3 | 0.83 ± 0.10 |
| **Rat heart** | 37.1 ± 1.1* | 50.5 ± 0.5 *** | 11.1 ± 0.8 *** | 1.20 ± 0.03* |

The mitochondrial Mi-CK and the three cytosolic MM-, MB-, and BB-CK isoforms in cardiac homogenates from 4 mice and 4 rats were separated by electrophoresis, and the fraction of each isoform is shown as mean ± SEM.

* denotes $p < 0.05$ and *** $p < 0.001$, significant difference between species.

respiration rate was recorded with either substrates for complex I alone (GM; protocol named $CK_{ADP\_GM}$) or substrates for complexes I and II (GMPS; protocol named $CK_{ADP\_GMPS}$). AK-stimulated respiration was recorded only with substrates for complexes I and II (GMPS; protocol named $AK_{ADP\_GMPS}$). A representative experimental trace of

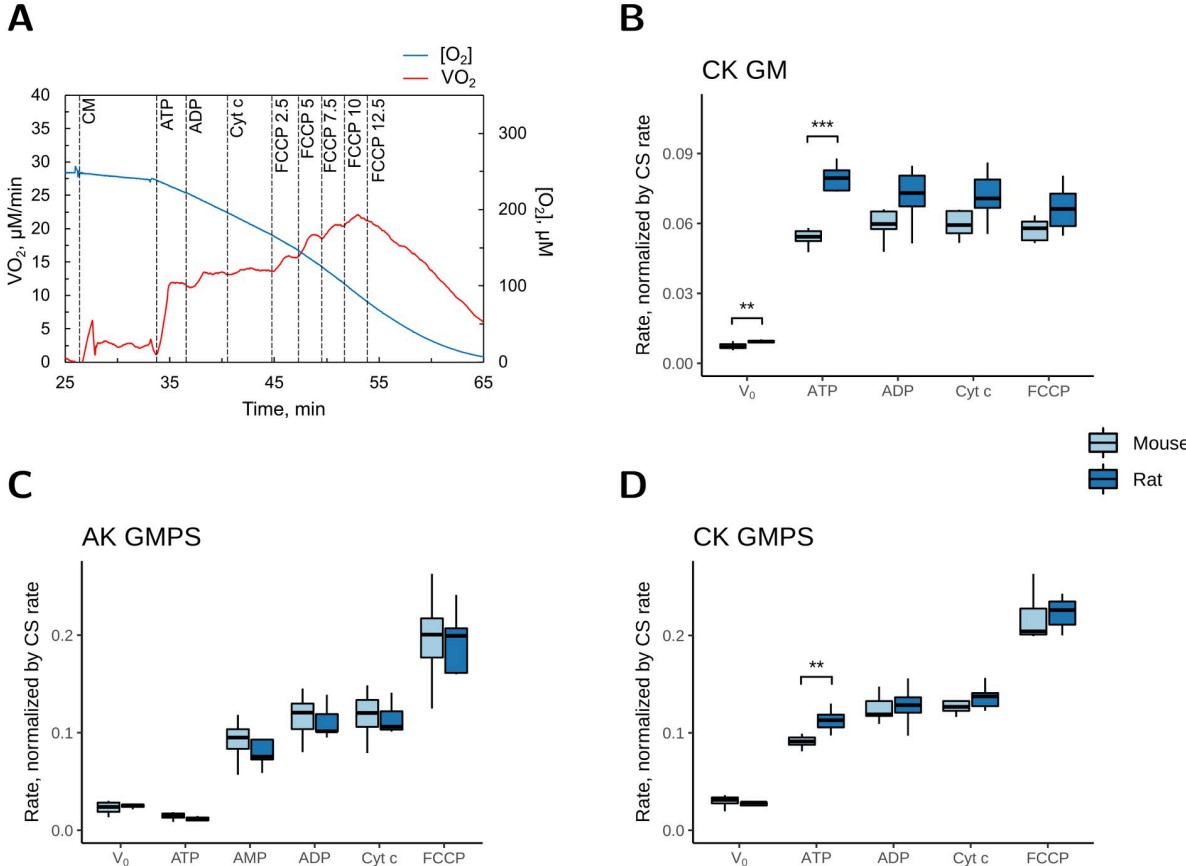

**Fig 3. Stimulation of respiration by CK and AK assessed relative to the maximal respiration rate in permeabilized mouse and rat cardiomyocytes.** When normalized to the CS activity, CK stimulated the respiration rate ($\mu mol\ O_2 \cdot min^{-1} \cdot IU\ CS^{-1}$) more in rat than in mouse cardiomyocytes. *A*: Representative example of a respirometer recording of $CK_{ADP\_GMPS}$ with permeabilized rat cardiomyocytes after addition of: CM–cardiomyocytes; ATP– 2 mM ATP; ADP– 2 mM ADP; Cyt c– 8 μM cytochrome c; FCCP–FCCP was gradually increased, the number behind shows the final concentration in μM. *B*: The averaged results of the experiments in the presence of GM, where respiration was stimulated by CK. *C* and *D*: The averaged results of the experiments in the presence of GMPS, where respiration was stimulated by AK (*C*) and CK (*D*). $V_0$ was subtracted from the subsequent rates. In B and D, as creatine was already present in the respiration chamber, ATP is the CK-stimulated respiration rate. In *C*, ATP is the rate stimulated by non-specific ATPases, and AMP is the AK-stimulated respiration rate. ADP is the maximal respiration rate recorded with 2 mM ADP. The number of animals was $n = 8$ and $n = 5$–7 for mice and rats, respectively. * denotes $p < 0.05$, ** $p < 0.01$, *** $p < 0.001$ significant difference between species.

$CK_{ADP\_GMPS}$ is shown in Fig 3A. Due to the low solubility of creatine, it was present in the medium before the start of the experiments at a concentration of 20 mM. After addition of cardiomyocytes, and recording of the basal respiration rate ($V_0$), the CK reaction was stimulated by the addition of 2 mM ATP. Then, 2 mM ADP was added to stimulate the respiration rate to $V_{O2\_max}$. Finally, cytochrome c (Cyt c) was added to test the intactness of the outer mitochondrial membrane, and the ionophore carbonyl cyanide-p-trifluoromethoxyphenyl-hydrazone (FCCP) was added in steps to determine the maximal electron flux capacity, i.e. the maximal uncoupled respiration rate. For AK-stimulated respiration measurements, AMP was added (instead of creatine) after addition of ATP. Fig 3B–3D show the averaged results and the statistical analyses from the respiration measurements of $CK_{ADP\_GM}$, $AK_{ADP\_GMPS}$, and $CK_{ADP\_GMPS}$, respectively, normalized to the CS activity in cell suspensions from mice and rats. Table 4 shows the respiration rates stimulated by CK ($V_{O2\_CK}$) and AK ($V_{O2\_AK}$), and the maximal coupled respiration rate ($V_{O2\_max}$) normalized to the CS activity of the cell suspension. In addition, it shows $V_{O2\_CK}$ and $V_{O2\_AK}$ relative to $V_{O2\_max}$. Note that in Fig 3 and Table 4, all respiration rates recorded after $V_0$ had $V_0$ subtracted before analysis. The same notation was used throughout the study.

Mitochondrial respiration has a lower phosphate to oxygen ($P/O_2$) ratio with GMPS than with GM, i.e. with GMPS as substrates, fewer ADP molecules are phosphorylated to ATP for a given $O_2$ consumption. This is because complex I substrates translocate 10 $H^+/O_2$, whereas complex II substrates translocate 6 $H^+/O_2$ across the inner mitochondrial membrane [42]. As the proton gradient across the inner mitochondrial membrane is the driving force for oxidative phosphorylation by the $F_1F_0$ ATPase, the higher $H^+/O_2$ with GM than with GMPS II leads to the theoretical $P/O_2$ ratios of 6 and 4 for GM and GMPS, respectively [43]. To determine whether differences in respiration rates were associated with differences in ADP-phosphorylation rates, we multiplied the respiration rates in Table 4 with these $P/O_2$ ratios to calculate the rates of ADP-phosphorylation, when respiration was stimulated by CK ($V_{ADP\_CK}$), AK ($V_{ADP\_AK}$), or 2 mM ADP ($V_{ADP\_max}$). These values are shown in Table 5.

**Table 4. Respiration rates of permeabilized mouse and rat cardiomyocytes stimulated by CK, AK, or 2 mM ADP.**

| | | $V_{O2\_CK}$ | $V_{O2\_AK}$ | $V_{O2\_max}$ | $V_{O2\_CK}/V_{O2\_max}$ | $V_{O2\_AK}/V_{O2\_max}$ |
|---|---|---|---|---|---|---|
| | | nmol $O_2$ /min/IU CS | | | % | |
| Mouse | GM | 55 ± 2 [##] | | 61 ± 3 | 91 ± 2 | |
| | GMPS | 93 ± 4 [####] | 89 ± 9 [####] | 124 ± 4 | 75 ± 1 | 79 ± 2 |
| Rat | GM | 77 ± 4 | | 72 ± 4 | 107 ± 4 | |
| | GMPS | 113 ± 4 [##] | 79 ± 7 [###] | 128 ± 7 | 89 ± 3 | 71 ± 3 |
| Substrate | | **** | | **** | *** | |
| Species | | *** | | | *** | * |

The respiration of permeabilized cardiomyocytes was recorded in the presence of either GM or GMPS as substrates. Under these conditions, the respiration rate is limited by the availability of ADP. The respiration was stimulated by endogenous ADP generated by CK ($V_{O2\_CK}$), or AK ($V_{O2\_AK}$). The maximal respiration rate ($V_{O2\_max}$), was recorded in the presence of 2 mM exogenous ADP. In addition, the fractional stimulation of respiration by CK and AK relative to $V_{O2\_max}$ was determined ($V_{O2\_CK}/V_{O2\_max}$ and $V_{O2\_AK}/V_{O2\_max}$, respectively). Values from 8 mice and 5–7 rats are shown as mean ± SEM.

[##] denotes $p < 0.01$,

[###] $p < 0.001$,

[####] $p < 0.0001$, significantly different from $V_{O2\_max}$.

* denotes $p < 0.05$,

*** $p < 0.001$,

**** $p < 0.0001$, significant effect of substrate or species. We found no interaction between substrates and species.

**Table 5. Estimated rates of ADP-phosphorylation by mitochondria stimulated by CK, AK or 2 mM ADP.**

| | | $V_{ADP\_CK}$ | $V_{ADP\_AK}$ | $V_{ADP\_max}$ |
|---|---|---|---|---|
| | | | nmol ADP/min/IU CS | |
| Mouse | GM | 332 ± 13 | | 367 ± 19 |
| | GMPS | 373 ± 14 | 354 ± 41 | 498 ± 18 |
| Rat | GM | 461 ± 23 | | 433 ± 27 |
| | GMPS | 451 ± 17 | 315 ± 26 | 512 ± 28 |
| Substrate | | | | **** |
| Species | | *** | | |

We used the data from Table 4 to estimate the rates of ADP-phosphorylation by mitochondria, when respiration was stimulated by endogenous ADP from CK ($V_{ADP\_CK}$) or AK ($V_{ADP\_AK}$), or 2 mM exogenous ADP ($V_{ADP\_max}$). The respiration rates were multiplied by a P/$O_2$ ratio of 6 or 4, for GM and GMPS, respectively, as explained in the main text. Values from 8 mice and 5–7 rats are shown as mean ± SEM.

*** denotes $p < 0.001$,

**** $p < 0.0001$, significant effect of substrate or species. We found no interaction between substrates and species.

The CK-stimulated respiration rates, $V_{O2\_CK}$, were species dependent and significantly higher in rat than in mouse cardiomyocytes in both $CK_{ADP\_GM}$ and $CK_{ADP\_GMPS}$ (Fig 3B and 3D, $p < 0.01$ and $p < 0.001$, respectively). However, after addition of ADP, Cyt c and FCCP, there was no significant difference in respiration rates between mice and rats (Fig 3B and 3D). As a result, the CK-stimulated respiration rate relative to the maximal respiration rate, $V_{O2\_CK}/V_{O2\_max}$, was higher in rats than in mice (Table 4; $p < 0.001$).

The CK-stimulated respiration rates were also substrate dependent (Table 4, compare Fig 3B and 3D), but the estimated rates of ADP-phosphorylation ($V_{ADP\_CK}$) were only affected by species and not substrate (Table 5). This suggests that the rate with which CK generated ADP to stimulate respiration was not affected by substrates but resulted in different $V_{O2\_CK}$ rates because of the substrate dependent P/$O_2$ ratios.

The respiration rate stimulated by AK, $V_{O2\_AK}$, did not differ between mouse and rat cardiomyocytes ($AK_{ADP\_GMPS}$; Fig 3C). However, when assessed relative to the maximal respiration, $V_{O2\_AK}/V_{O2\_max}$ was slightly higher in mouse than in rat cardiomyocytes (Table 4; $p < 0.05$).

When $V_{O2\_max}$ was normalized to the CS activity of the cell suspension, there was no difference between mouse and rat cardiomyocytes (Fig 3 and Table 4). However, the CS activity was higher in mouse than in rat cardiomyocytes (Table 2). When $V_{O2\_max}$ was normalized to the protein content of the cell suspension (see S1 Table), it was significantly higher in mouse than in rat cardiomyocytes. Thus, $V_{O2\_max}$ correlated with the CS activity of the cell suspension.

$V_{O2\_max}$ was significantly higher than $V_{O2\_CK}$ except in rat cardiomyocytes with only complex I substrates (GM). $V_{O2\_max}$ was also significantly higher than $V_{O2\_AK}$ in both mouse and rat cardiomyocytes (Table 4).

$V_{O2\_max}$ was significantly affected by the substrates. In the presence of GM, $V_{O2\_max}$ was lower than in the presence of GMPS (Table 4). $V_{ADP\_max}$ was also significantly affected by the substrates (Table 5). Thus, in the presence of 2 mM ADP, the rate of ADP-phosphorylation was lower with GM alone than with GMPS.

The quality of the permeabilized cardiomyocytes was assessed through the coupling efficiency of respiration and the Cyt c test. The coupling efficiency of respiration, calculated as $1-V_0/V_{O2\_max}$, indicates the proportion of oxygen used for ATP synthesis. In permeabilized cardiomyocytes, a high coupling efficiency indicates that they are 1) adequately permeabilized

and respond to ADP addition, and 2) not damaged, as this would cause an elevated $V_0$. According to Chance and Williams, tightly coupled mitochondria show a substrate-dependent 4- to 10-fold increase in respiration rate upon addition of ADP [44]. This corresponds to a coupling efficiency of 0.75–0.90. In the present experiment, the coupling efficiency was similar in mouse and rat cardiomyocytes but substrate dependent and lower with GMPS than with GM ($81.6 \pm 0.5\%$ and $89.7 \pm 0.7\%$, respectively; $p < 0.0001$; data were pooled for rats and mice).

The addition of Cyt c is commonly used to test the intactness of the outer mitochondrial membrane. If the outer mitochondrial membrane is damaged, electron transfer will be compromised as Cyt c leaks out of the mitochondria. The associated decline in respiration rate will be rescued upon addition of Cyt c. In our experiments, the addition of Cyt c did not significantly affect the respiration rate in any of the measurements (Fig 3), indicating that in the cardiomyocytes in the present study, the outer mitochondrial membrane was intact.

The addition of FCCP, an uncoupler of respiration, did not increase the respiration rate with GM only. But with GMPS, FCCP increased the respiration rate by ~40% (Fig 3). The phosphorylation control ratio, calculated as $V_{O2\_max}/V_{FCCP}$, was $106 \pm 2.1$ and $58.3 \pm 0.6\%$ with GM and GMPS, respectively (pooled data from rats and mice).

## Channelling of ADP from CK or AK to the mitochondria

The ADP channelling between CK or AK and the mitochondria was assessed in parallel experiments, where the kinase stimulated respiration was subsequently inhibited by addition of PEP and PK. Using the same substrate-kinase combinations as before, CK-stimulated respiration rate was recorded with either GM ($CK_{PEP/PK\_GM}$) or GMPS ($CK_{PEP/PK\_GMPS}$), and AK-stimulated respiration was recorded only with GMPS ($AK_{PEP/PK\_GMPS}$). Fig 4A shows a representative experimental trace from a recording of $CK_{PEP/PK\_GMPS}$. After addition of cardiomyocytes (CM) to the respiration chamber, addition of ATP to the solution (already containing 20 mM creatine) stimulated CK to generate endogenous ADP. This ADP distributed in the solution and stimulated respiration. Then, endogenous PK was stimulated by addition of PEP and competed with mitochondrial respiration for some of the ADP. Subsequently, exogenous PK in excess was added to the chamber. Thus, PK converted all accessible ADP in the solution to ATP, leaving only ADP that was directly channelled from CK to mitochondria to stimulate respiration. For AK-stimulated respiration measurements, AMP was added (instead of creatine) after addition of ATP. Fig 4B–4D, shows the averaged results and the statistical analysis from the respiration measurements of $CK_{PEP/PK\_GM}$, $AK_{PEP/PK\_GMPS}$, and $CK_{PEP/PK\_GMPS}$, respectively, normalized to the CS activity in cell suspensions from mouse and rat hearts.

CK-stimulated respiration was significantly higher in rats than in mice irrespective of the substrates (Fig 4B; $p < 0.0001$; Fig 4D; $p < 0.01$), in agreement with the results in Figs 1B and 3. In $CK_{PEP/PK\_GM}$, the addition of PEP and PK lowered respiration rate more in rat than in mouse cardiomyocytes (by $75 \pm 1\%$ and $58 \pm 1\%$, respectively; $p < 0.001$), and the respiration rate in the presence of PEP and PK was lower in rat than in mouse (Fig 4B). In $CK_{PEP/PK\_GMPS}$, the addition of PEP and PK also lowered respiration rate more in rat than in mouse cardiomyocytes (by $62 \pm 1\%$ and $48 \pm 1\%$, respectively; $p < 0.001$), as the initially higher respiration rate in rat cardiomyocytes was lowered to the same level as in mice in the presence of PEP and PK (Fig 4D).

In $AK_{PEP/PK\_GMPS}$, there was no difference between mice and rats, and the respiration rate was lowered to the same level as before addition of AMP (Fig 4C; compare rates at ATP and PK).

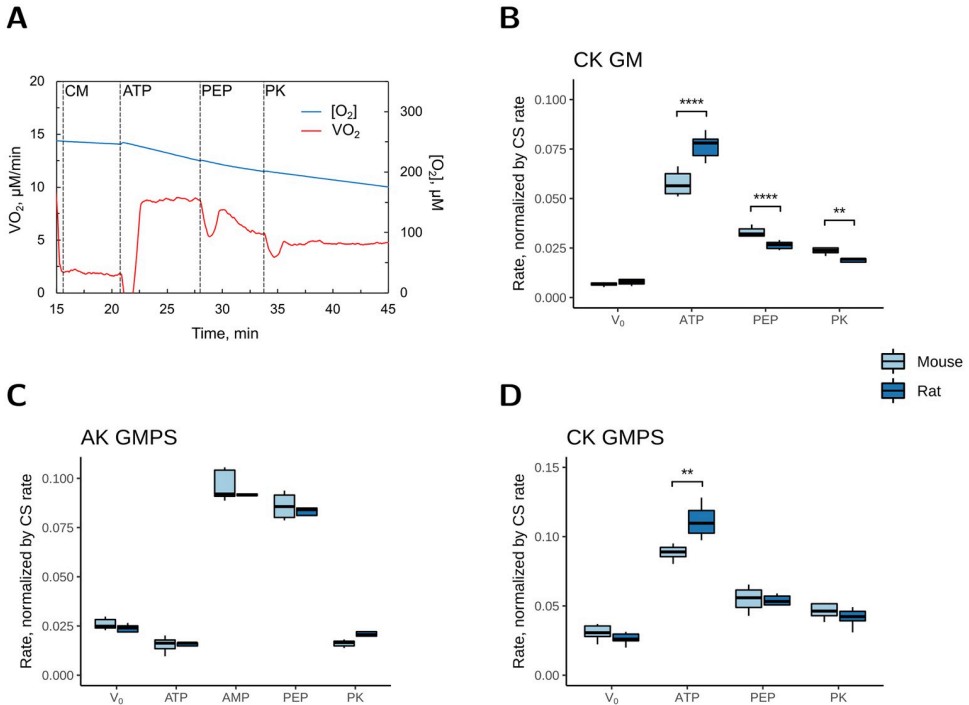

**Fig 4. Channeling of ADP from creatine kinase (CK) and adenylate kinase (AK) to mitochondria in permeabilized mouse and rat cardiomyocytes.** As in Fig 3, when normalized to the CS activity, CK stimulated the respiration rate ($\mu$mol $O_2 \cdot min^{-1} \cdot IU$ $CS^{-1}$) more in rat than in mouse cardiomyocytes. However, this species difference was reversed (B) or lost (D) after addition of PEP and PK to compete with the mitochondria for endogenous ADP from CK. *A*: Representative example of a respirometer recording of $CK_{PEP/PK\_GMPS}$ with permeabilized rat cardiomyocytes after addition of: CM–cardiomyocytes; ATP– 2 mM ATP; PEP– 5 mM PEP; PK– 20 U/ml PK. *B*: The averaged results of the experiments in the presence of GM, where respiration was stimulated by CK. *C* and *D*: The averaged results of the respiration experiments in the presence of GMPS, where respiration was stimulated by AK (*C*) and CK (*D*). $V_0$ was subtracted from the subsequent rates. For further explanation, see the legend of Fig 3 and the main text. The number of animals was $n = 7–8$ and $n = 5–7$ for mice and rats, respectively. * denotes $p < 0.05$, ** $p < 0.01$, *** $p < 0.001$, **** $p < 0.0001$ significant difference between species.

## Discussion

This is the first study to show that there are differences between rat and mouse hearts in terms of both aerobic capacity, CK isoform distribution, and intracellular compartmentalization. When comparing the measurements in homogenate and permeabilized cardiomyocytes, we were surprised to find that CK and, to an even greater extent, AK activities assessed in heart homogenates were much higher than the ADP-phosphorylation rates estimated from the CK- and AK-stimulated respiration rates of permeabilized cardiomyocytes. This is a consequence of intracellular compartmentalization and demonstrates that results from whole-heart homogenates cannot be directly extrapolated to the situation in permeabilized cardiomyocytes.

### Rat hearts have a lower oxidative capacity than mouse hearts

The CS activities reported in the present study were close to activities reported in other studies [35, 45–48]. The CS activity was higher in mouse than in rat in both whole heart homogenates and cell suspensions (Fig 1A and Table 2). Furthermore, when normalized to protein content, $V_{O2\_max}$ was higher in mouse than in rat cardiomyocytes (S1 Table). Thus, all three measurements indicate that rat hearts have a lower oxidative capacity than mouse hearts.

In skeletal muscle, the oxidative capacity correlates with the CO activity [34]. However, in the present study, the CO activity was similar in mouse and rat hearts (Fig 1A), and $V_{O2\_max}$ correlated with the CS activity (Fig 3). This suggests that, in heart muscle, the CS activity is a better marker of oxidative capacity.

As the CS activity correlated with the mitochondrial oxidative capacity, and one of our aims was to assess the rate of ADP-generation by AK and CK relative to the mitochondrial oxidative capacity, we normalized the respiration data to the CS activity. As described in the Introduction, we assumed that CS is mainly from cardiomyocytes, and that the CS in other cell types is negligible. Furthermore, we assumed that this distribution of CS is similar in rat and mouse heart, so that normalizing to CS allowed us to compare the reaction rates in whole heart homogenates and isolated cardiomyocytes across the species.

## Rat hearts have more Mi-CK and B-CK, and mouse hearts have more M-CK

The CK isoform distribution was also different in mouse and rat hearts (Fig 2 and Table 3). Mouse hearts had a larger fraction of MM-CK, whereas rat hearts had larger fractions of Mi-, MB- and BB-CK. The large fraction of MB-CK in rat heart is consistent with results in the literature [14, 17, 49]. We are uncertain whether rat heart benefits functionally from having a greater expression of B-CK. B-CK has a higher creatine affinity than M-CK [50], but the total creatine content seems to be similar in rat and mouse ventricles (~80 nmol/mg protein) [49, 51]. MM-CK and BB-CK are both mainly soluble, but a fraction associates to cellular structures, and they seem to do so differently. Their N-terminal regions differ, and due to four conserved lysine residues, MM-CK binds to the M-band, whereas BB-CK does not [52, 53]. Instead, B-CK binds to the I-band of the myofibrils [52, 54, 55]. MM-CK is also found at the I-band of the myofibrils, but here, it is bound loosely and through phosphofructokinase of the glycolytic pathway [56]. Both MM-CK and BB-CK are known to associate with membranes. In muscle tissue, MM-CK associates near the sarcoendoplasmic reticulum $Ca^{2+}$-ATPase (SERCA) and the $Na^+/K^+$-ATPase [57, 58]. More recent evidence suggests that BB-CK also locates near membrane structures, in some cases in a manner that is regulated through phosphorylation by AMP-activated protein kinase (AMPK), suggesting that this association may be weaker and transient depending on the state of the cell [59]. We were unable to find information regarding the heterodimeric MB-CK. At present, we speculate whether the differences in CK isoform composition in rat and mouse heart relate to the different binding properties of M- and B-CK, but this warrants further studies.

## Rat hearts have a higher CK activity and larger fraction of Mi-CK, but less ADP from CK is channelled to the mitochondria

Traditionally, the reverse CK activity is recorded in homogenates. When taking into account temperature differences, the CK activities in rat and mouse hearts (Fig 1A) were close to and a little higher, respectively, than reported in other studies [45, 46, 48, 60]. When normalized to the wet weight, $CK_r$ tended to be higher and $CK_f$ activity was slightly, but significantly higher in rat than in mouse heart (Fig 1A).

When normalized to the CS activity, the CK activities were clearly higher in rat than in mouse heart (Fig 1B). This difference was also reflected in the experiments on permeabilized cardiomyocytes, where stimulation of CK led to higher $V_{O2\_CK}$ in rat than in mouse cardiomyocytes (Fig 3B and 3D and Table 4). As expected, $V_{O2\_max}$ was substrate dependent with $V_{O2\_max\_GM}$ being lower than $V_{O2\_max\_GMPS}$ (compare Fig 3B and 3D; Table 4). This has also been shown before [35, 36]. As a result, $V_{O2\_CK}/V_{O2\_max}$ was also substrate dependent

(Table 4; effect of substrate), and consistently higher in rat than in mouse heart (Table 4; effect of species).

Although rat cardiomyocytes had a higher $V_{O2\_CK}/V_{O2\_max}$, the subsequent addition of PEP and PK to competitively inhibit the flux of ADP from CK to mitochondria lowered the respiration rate more in rat than in mouse cardiomyocytes. In the presence of PEP and PK, the respiration rate was similar ($CK_{PEP/PK\_GMPS}$) or slightly lower ($CK_{PEP/PK\_GM}$) in rats than in mice (Fig 4B and 4D). This suggests that in mouse hearts, a larger fraction of ADP generated by CK was channelled to the mitochondria. However, the fraction of Mi-CK was lower in mouse than in rat hearts (Fig 2 and Table 3). It may seem contradictory that mouse cardiomyocytes have less Mi-CK and greater ADP channelling to the mitochondria. However, this can be explained by differences in the intracellular compartmentalization. The greater channelling of ADP from CK to mitochondria in mouse heart could be due to a tighter coupling of cytosolic CK as well as Mi-CK to the respiration.

As to Mi-CK, in isolated mitochondria, approximately half of the ADP generated by Mi-CK is translocated by the ANT into the matrix, while the rest diffuses out through the voltage dependent anion channel (VDAC) in the outer mitochondrial membrane [6, 61]. *In vivo* and in permeabilized cardiomyocytes, the mitochondrial outer membrane permeability is more restricted [62]. This, in turn, is expected to increase the channeling between Mi-CK and ANT. Thus, it is possible that there is more direct transfer between Mi-CK and ANT or the outer mitochondrial membrane is less permeable to ADP in mouse cardiomyocytes, so more ADP from Mi-CK cycles within the mitochondria and is inaccessible to PK.

As to cytosolic CK, it was shown in Mi-CK knockout mice that cytosolic CK can also be coupled to respiration [63] possibly due to intracellular diffusion barriers in the cytosol, which can group CK and mitochondria [39, 64–67]. The extent of channelling we observe between cytosolic CK and the mitochondria depends on the interplay between cytosolic CK, PK, and diffusion barriers. It is possible that in mouse cardiomyocytes, a larger fraction of cytosolic CK is on the mitochondrial side of the diffusion barriers, or the diffusion barriers are less permeable, so more ADP cycles between CK and the mitochondria and is inaccessible to PK.

If the cytosolic diffusion barriers are less permeable to ADP, then they are presumably also less permeable to the diffusion of proteins. This is a relevant point for the present experiments on permeabilized cardiomyocytes, which do not necessarily reflect the situation *in vivo*, because some cytosolic CK and PK may diffuse out of the cells, and also some exogenous PK diffuses into the cell. The diffusion of CK out of the cardiomyocytes does not on its own inhibit respiration, because CK continues to generate ADP in the solution. However, it may lead to an underestimation of the overall coupling between all CK isoforms and respiration. Thus, if the cytosolic diffusion barriers are less permeable in mouse cardiomyocytes, it is also possible that the species differences in the PEP-PK assay are caused in part by the differences in CK and PK diffusion between solution and cardiomyocytes. Clearly, further studies are needed to pinpoint the exact mechanism behind the different outcomes of the PEP-PK assay. Nevertheless, our results suggest that compartmentalization and/or energy transfer is different in rat and mouse cardiomyocytes.

The lowering of respiration rate by PEP and PK in mouse cardiomyocytes is in agreement with our previous finding [21]. With the present study, we extend this finding to rat cardiomyocytes. However, our results contradict the findings from another group, who found on rat cardiomyocytes that with GM as substrates CK-stimulated respiration rate was similar to $V_{O2\_max}$ even in the presence of PEP and PK [22, 23]. We speculated whether this difference between studies could be because the cardiomyocytes in the other study had damaged mitochondria. If the mitochondria are damaged, $V_{O2\_max}$ will be very low and $V_{O2\_CK}/V_{O2\_max}$ will be high. In order to compare our data with those of others, we also normalized our maximal

respiration rate in rat cardiomyocytes to the Cyt $aa_3$ content (S1 Table, $V_{O_2\_max}$/cyt $aa_3$). In the present study, $V_{O_2\_max\_GM}$ was ~35% higher than reported by others (273 ± 19 versus 178 ± 34 nmol $O_2 \cdot min^{-1}$nmol ·cyt $aa_3^{-1}$) [68]. However, in rat cardiomyocytes in the present study, the addition of PEP and PK lowered the respiration rate by 75% (Fig 4B). As the difference in $V_{O_2\_max}$ between studies was smaller than the lowering of respiration rate by PEP and PK, the higher $V_{O_2\_max}$ in the present study can only partially explain the difference between the studies. In the present study, the cell viability was acceptable (Table 2), the addition of Cyt c had no effect on respiration rate (Fig 3B–3D), and the coupling efficiency of respiration was high. Therefore, we are confident that our preparation was sound.

Taken together, our study confirms that intracellular compartmentalization leads to a local pool of phosphates circulating between mitochondria and CK and suggests that this is more predominant in mouse than in rat cardiomyocytes. It must be noted that these experiments were performed on isolated, non-contracting cardiomyocytes, and it is uncertain how deformation of the cells during contraction affects the compartmentalization.

### The rate of ADP-generation by CK is lower than the maximal rate of ADP-consumption by the mitochondria

As noted above, rat hearts had a higher $V_{O_2\_CK}$/$V_{O_2\_max}$ than mouse hearts. According to our data, mitochondrial and cytosolic CK stimulated respiration rate to 90% of the maximal rate with GMPS in rat (Fig 3D and Table 4). This is far from the finding on isolated rat hearts that the CK reaction rate is 10 times higher than the maximal respiration rate [4, 5], but it is in agreement with a previous study [69]. This may explain why the CK shuttle is bypassed under extreme workloads [69], and why studies on transgenic mice with disturbances in the CK system have been equivocal regarding the importance of the CK system [11].

### Rat and mouse hearts have similar $V_{O_2\_AK}$/$V_{O_2\_max}$

In homogenates, the AK activity normalized to the wet weight was ~30% higher in mouse than in rat hearts, measured in both directions (Fig 1A). In mouse hearts, the AK activity was ~1.5–2 times higher than reported previously [21, 45, 47], whereas in rat hearts, it was only slightly higher than in another study [46]. In contrast to our previous study on creatine-deficient mice, where AK had the highest activity of the kinases [21], the present results showed that the AK activity was lower than the CK activity in both mouse and rat hearts, when measured in the direction of ATP-production. The AK activities were similar in the forward and reverse directions, which is compatible with another study [70], showing that the $AK_f$/$AK_r$ ratio is ~1.3.

When the AK activity was normalized to the CS activity in homogenates, it was similar in mouse and rat heart (Fig 1B). In agreement with this, there was no difference between rat and mouse in the respiration experiments (Fig 3C). However, relative to the maximal respiration rate, $V_{O_2\_AK}$/$V_{O_2\_max}$ was slightly higher in mouse than in rat hearts (Table 4), but below 80%. In our previous experiments, AK stimulated respiration to the maximum, but they were performed with only GM as substrates [21]. In the present study, AK-stimulated respiration measurements were performed only with GMPS, and as $V_{O_2\_max}$ is higher with GMPS than with GM, it was not surprising, that $V_{O_2\_AK}$/$V_{O_2\_max}$ was lower than recorded with GM, as was also the case for CK (Fig 3B and 3D, Table 4).

### No channelling of ADP from AK to the mitochondria

When the flux of ADP from AK to the mitochondria was inhibited by the subsequent addition of PEP and PK, the respiration rate was lowered to the same level as before the addition of AMP in both rat and mouse cardiomyocytes (Fig 4C). This was similar to our previous results

on mouse cardiomyocytes [21], and in agreement with the low expression of AK2 in the mouse heart [32], but in contrast to another study on isolated rat heart mitochondria [27]. We hypothesize that methodological differences could cause this discrepancy between studies. The present study suggests that the majority of AK is cytosolic in both rat and mouse hearts.

## AK and CK activities in homogenate are higher than estimated $V_{ADP\_AK}$ and $V_{ADP\_CK}$ in permeabilized cardiomyocytes

We estimated the rates of ADP-generation by CK and AK in permeabilized cardiomyocytes ($V_{ADP\_CK}$ and $V_{ADP\_AK}$, Table 5) by multiplying the respiration rates with the $P/O_2$ ratios (4 for GMPS and 6 for GM). This represents their forward reaction rates, and we had expected that they would be similar to the forward reaction rates recorded in homogenate ($CK_f$ and $AK_f$, Fig 1B). Surprisingly, we found that the rates estimated from permeabilized cardiomyocytes were much lower than in homogenate. $CK_f$ was ~2 times higher and $AK_f$ was ~9 times higher than $V_{ADP\_CK}$ and $V_{ADP\_AK}$, respectively. This difference is highlighted in Fig 5. Our finding suggests that in permeabilized cardiomyocytes with the intracellular structures left intact, local substrate and product concentrations in the vicinity of an enzyme can be very different from the concentrations in solution. One factor is that diffusion of substrates into the permeabilized cardiomyocytes is restricted so the substrate concentrations could be smaller

### Cell suspension

### Tissue homogenate

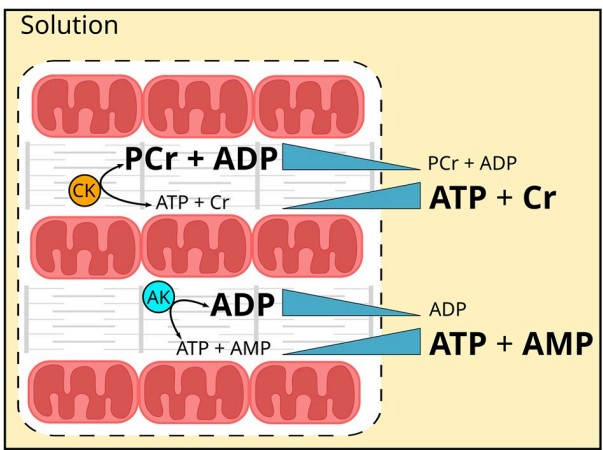
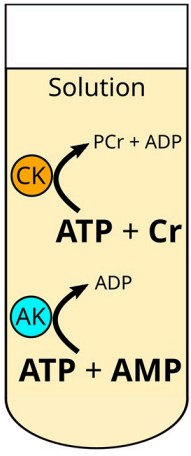

| | Estimated rate of ADP-generation µmol ADP/min·IU CS | | Enzyme activity µmol ADP/min·IU CS | |
|---|---|---|---|---|
| | Mouse | Rat | Mouse | Rat |
| **CK** | 0.37 | 0.45 | 0.59 | 0.91 |
| **AK** | 0.35 | 0.31 | 2.97 | 2.80 |

**Fig 5. Intracellular compartmentalization shapes energy transfer in cardiomyocytes.** In permeabilized cardiomyocytes (left panel; data from Table 5, units converted to µmol ADP·min$^{-1}$·IU CS$^{-1}$), diffusion of substrates to the center of the cell is restricted. Thus, kinases in the center of the cell may be exposed to smaller concentrations of substrates than are present in the surrounding solution. More importantly, as diffusion out of the cell is restricted, the products accumulate near the kinase and inhibit the reaction rate. In contrast, when recording the kinase activity in homogenate (right panel; data from Fig 1), the kinases are in solution, where diffusion is much faster. Thus, there is no build-up of products, and the reaction takes place without inhibition.

than in solution. More importantly, the diffusion of products out of the permeabilized cardiomyocytes is restricted, so the products accumulate near the kinases, inhibiting the reaction rate. We have illustrated this in Fig 5.

The restriction of diffusion in cardiomyocytes has been studied for several decades. In cardiomyocytes, intracellular membranes such as the transverse tubules, the sarcoplasmic reticulum, and the mitochondria, together with protein dense parts of the sarcomeres constitute barriers that form modules [64, 65]. These modules influence energy transfer within the cell [39]. Local environments are also known to play a crucial role in cAMP signalling [71], excitation-contraction coupling [72] and mitochondrial calcium uptake [73, 74]. In the present study, we were surprised to see that compartmentalization had such a large effect on the AK activity. This, in turn, is likely to affect not only ATPase function, but also energetic signalling through, for example, AMPK, which is a cellular energy sensor implicated in both acute signalling and regulation of gene expression [75]. Our results highlight the challenge for computational models, which should take into account the intracellular heterogeneity of substrate and product concentrations and the spatial limitations of their diffusion inside the cell [62, 64, 65].

## Conclusions

In the present study, we found species differences between mouse and rat hearts. Rat hearts had a lower oxidative capacity than mouse hearts. As a result, CK/CS and $V_{O2\_CK}/V_{O2\_max}$ were higher in rat than in mouse, and the distribution of CK isoforms was different. In rat heart, although $V_{O2\_CK}/V_{O2\_max}$ and the fraction of Mi-CK was higher than in mouse heart, less ADP was channelled from CK to the mitochondria. This suggests differences in the compartmentalization of mouse and rat cardiomyocytes.

An interesting finding of this study was that AK/CS activity in whole tissue homogenates was several times higher than the $V_{ADP\_AK}$ estimated from the respiration rate in isolated permeabilized cardiomyocytes. This difference is a consequence of intracellular compartmentalization. Our results highlight how intracellular structural organization shapes energetic compartmentalization, which plays a pivotal role in energy homeostasis, signalling, and regulation of cardiac metabolism.

## Materials and methods

All experiments and animal procedures complied with directive 2010/63/EU of the European Parliament for the protection of animals used for scientific purposes and were approved by the Project Authorisation Committee for Animal Experiments in the Estonian Ministry of Rural Affairs. All methods are reported in accordance with ARRIVE guidelines.

### Animals

The animals used in this study were 7–10 months for mice and 10–12 months for rats. Due to the low quality of cardiomyocytes isolated from male rat hearts, only females were used in this study. Sprague-Dawley rats were a gift from the Laboratory of Neurobiology at Tallinn University of Technology. C57BL/6J Ola Hsd mice were originally from Envigo RMS B.V. (The Netherlands). The animals were kept in the animal facility of Tallinn University of Technology at an ambient temperature of 22–22.8°C and a 12:12 hours light:dark cycle. They had free access to water and food (V1534-000 Rat/mouse maintenance from Ssniff Spezialdiäten GmbH, Germany).

## Isolation of cardiomyocytes

Cardiomyocytes from mouse [15] and rat [76] hearts were isolated using a slightly modified version of a method described previously. The mice were anesthetized with a mixture of ketamine/dexmedetomidine (150 mg/kg and 0.5mg/kg, respectively) and received an injection of 250U of heparin to prevent blood coagulation. When the toe-pinch reflex was absent, the animal was euthanized by cervical dislocation. The rats were anesthetized with 2% isoflurane using Open-Drop system (or Drop Jar method) (https://animal.research.uiowa.edu/iacuc-guidelines-anesthesia) [77]. Briefly, the rats were placed in a closed container of known volume (~5l) with tightly fitting lid, a gauze pad soaked with appropriate volume (~0.5 ml) of isoflurane was placed in the bottom of the container and the animals were left to inhale isoflurane vapours. When they lost the righting reflex and breathing had slowed but was regular, 2500U of heparin was injected intraperitoneally and they were allowed to sleep a little bit more. Under deep isoflurane anaesthesia, the rats were euthanized by decapitation. The hearts of both rat and mouse were excised and immediately placed in ice-cold wash solution consisting of the following (mM): 117 NaCl, 5.7 KCl, 1.5 $KH_2PO_4$, 4.4 $NaHCO_3$, 1.7 $MgCl_2$, 21 HEPES, 20 taurine, 11.7 glucose, and 10 2,3-butanedione monoxime (pH was adjusted to 7.4 with NaOH). It was cannulated via the aorta on a Langendorff perfusion system. The heart was first perfused with wash solution at 38.5˚C at a constant pressure of 80 cm $H_2O$. When the heart was washed free of blood, the perfusion was switched to a constant flow with digestion solution containing 0.37–0.435 mg/ml Liberase DL (Roche) and 1.36 mg/ml of dispase II (Roche). The pressure was observed for 10–15 minutes or ~30 min (mouse and rat heart, respectively) until the pressure had decreased to 40–50% of the initial. When the heart was soft, the perfusion was stopped. The ventricles were cut into smaller pieces, transferred to a beaker with digestion solution and incubated further at 38.5˚C with gentle shaking until the tissue started falling apart. Cells were harvested with a Pasteur pipette several times and filtered through a 100μm cell strainer (EASYstrainerTM Cell Strainer, Greiner Bio-One) into a vial with sedimentation solution consisting of wash solution (without 2,3-butanedione monoxime) containing additional 2mM pyruvate, 10 μM leupeptin, 2 μM soybean trypsin inhibitor, and 3 mg·mL$^{-1}$ BSA. The viable cells were separated by sedimentation or by centrifugation for 2 min at 300 rpm/ 12g. During the first washes, extracellular $Ca^{2+}$ was gradually increased to 2 mM to ensure $Ca^{2+}$ tolerance of the cells. Then, extracellular $Ca^{2+}$ was washed out again by washing the cells three times with 5–8 ml of sedimentation solution. The isolated cells were stored in this solution at room temperature until use within 3 hours.

To assess the quality of the cell preparation, the yield of the cell suspension was measured with a 1000 or 5000 μl pipette (Eppendorf), and a 1:10 dilution of the cells was counted in a chamber to estimate the total number of cells as well as the viability (the percentage of live cells relative to the total number of cells).

## Respiration measurements

For the respiration experiments, we used protocols similar to those described previously [21, 35]. In brief, we used a Strathkelvin RC 650 Respirometer equipped with six 1302 O2-electrodes connected via a 929 Oxygen System interface (all from Strathkelvin Instruments Limited, UK). The respirometer was thermostatted to 25˚C (Julabo F12-ED, JULABO Labortechnik GmbH). The respiration measurements in cardiomyocytes were performed in 2 ml of respiration solution consisting of 110 mM sucrose, 60 mM K-lactobionic acid, 3 mM $KH_2PO_4$, 3 mM $MgCl_2$, 20 mM HEPES, 20 mM taurine, 0.5 mM EGTA, 0.5 mM dithiothreitol (DTT) (pH was adjusted to 7.1 with KOH). 5 mg·mL$^{-1}$ BSA and 25 μg·mL$^{-1}$ saponin were added just before use. Saponin was present in the respiration chamber throughout the

measurements. It interacts with cholesterol to form pores in the sarcolemma, which contains 90% of the cellular cholesterol [78]. At this concentration, saponin does not damage the mitochondrial membranes, which have a very low cholesterol content [79]. CK experiments were performed in the presence of 2.5 mM glutamate and 2 mM malate only (CKGM) and in the presence of 2.5 mM glutamate, 2 mM malate, 5 mM pyruvate and 15 mM succinate (CKGMPS). AK experiments were carried out only in the presence of 2.5 mM glutamate, 2 mM malate, 5 mM pyruvate and 15 mM succinate ($AK_{GMPS}$).

We used CK and AK protocols similar to those described in [21]. First, 5 ul of cell suspension, for CKGMPS and $AK_{GMPS}$, and 10 ul for CKGM were added to the respiration chambers. Cells were allowed at least 5 min to permeabilize before the steady-state basal respiration rate, $V_o$, was recorded. After that, we stimulated AK by adding 2 mM ATP and 1 mM AMP, or CK by adding 2 mM ATP while 20 mM creatine was already present in the respirometer chamber before the cells were added.

In the chambers, where we recorded $CK_{PEP/PK\_GM}$, $CK_{PEP/PK\_GMPS}$, and $AK_{PEP/PK\_GMPS}$, this was followed by the addition of 5 mM PEP and 20 U/ml exogenous PK.

In the parallel chambers, where we recorded $CK_{ADP\_GM}$, $CK_{ADP\_GMPS}$, and $AK_{ADP\_GMPS}$, this was followed by the addition of 2 mM ADP, 10 μM Cyt c and stepwise titration of FCCP in 2.5 μM steps until the maximum uncoupled respiration rate, $V_{FCCP}$, was reached.

As a reference, respiration rate measured at 2 mM ADP was taken as the maximal coupled respiration rate, $V_{O2\_max}$. This measurement was used to estimate AK and CK stimulated respiration rates ($V_{O2\_AK}$ and $V_{O2\_CK}$, respectively) relative to $V_{O2\_max}$ ($V_{O2\_CK}/V_{O2\_max}$ and $V_{O2\_AK}/V_{O2\_max}$), and to calculate the coupling efficiency of respiration as follows: $(V_{O2\_max}-V_0)/V_{O2\_max}$ [37]. This allowed us to determine the oxidative phosphorylation control ratio, defined as: $V_{O2\_max}/V_{FCCP}$ [37].

## Homogenization

Cardiac homogenates were prepared as in Barsunova et al. [80]. The mice and rats were anesthetized and killed as described above for the isolation of cardiomyocytes. The heart was quickly removed from animals and immediately transferred to a glass beaker with ice-cold isolation solution. The heart was trimmed of any obvious fat and connective tissue, gently blotted to remove excess fluids, weighed, cut into several pieces if needed (for rat heart), and then stored in cryovials at -80°C until further experiments. All subsequent homogenization procedures were carried out on ice. The heart tissue was minced with scissors into small pieces, transferred to a glass homogenizer, and ice-cold homogenization buffer was added to a concentration of 50 mg tissue/ml buffer. The buffer consisted of 5 mM HEPES, 1mM EGTA, 0.1% Triton X-100, 1 mM DL-Dithiothreitol, and 1 tablet of cOmplete Mini Protease inhibitors per 10 ml buffer (Roche, Merck) (pH 8.7). Next, the heart tissue was ground with a pestle attached to a drill until the solution was homogenous. The homogenized samples were incubated on ice for one hour before use. Fresh, non-diluted homogenates were use to measure CO activity. The remaining homogenates were kept at -80°C until activities of CS, CK and AK were measured.

## Enzyme activities

Enzyme activities were recorded using Evolution 600 spectrophotometer (Thermo Fisher Scientific) equipped with a Peltier water-cooled cell changer (SPE 8 W, Thermo Fisher Scientific) to maintain temperature at 25°C.

CO and CS activities were determined as described earlier [80]. CO activity was determined by measuring the decrease in absorbance, caused by oxidation of Cyt c by cytochrome

oxidase, at 550 nm [81]. The reaction took place in 1 ml 13 mM sodium phosphate buffer (pH 7.4) containing 0.4 mg/ml Cyt c, which had been reduced with Na-dithionite. After recording the initial absorbance for 10–20 seconds, the reaction was initiated by the addition of 10 µl of undiluted homogenate. The reaction of Cyt c oxidation represents a first-order reaction with respect to reduced Cyt c and is observed as a logarithmical decline in absorption (as a function of time). The rate constant, obtained by fitting to the equation $\delta$(Absorption) / $\delta$(time) = k (Absorption), was normalized to the tissue wet weight, $min^{-1} \cdot g^{-1}$. The IOCBIO Kinetics software for fitting is open source and available at https://iocbio.gitlab.io/kinetics.

CS activity was recorded using a coupled enzyme assay in a total volume of 1 mL CS buffer containing the following (in mM): 100 Tris·HCl buffer (pH 8.1), 0.1 5,5' -dithiobis(2-nitrobenzoic acid) (DTNB), and 0.3 acetyl-CoA. The assay was started by the addition of 10 µL or 15–20 µL of diluted (1:10) cell suspension or heart homogenate, respectively. The change in absorbance was recorded for 2 min at 412 nm before (for reference) and after addition of 0.5 mM oxaloacetate. The enzyme activity was calculated using the extinction coefficient for thionitrobenzoate (TNB), which is 14150 $M^{-1}$ $cm^{-1}$ at 25°C [82].

The activities of CK and AK in the reverse direction (ATP-production) were measured in a coupled enzyme assay in a total volume of 1ml respiration buffer (without BSA and saponin) (see composition in Respiration measurements section) containing the following: 10 mM glucose, 0.6 mM NADP, 2 mM ADP, 5 $U \cdot ml^{-1}$ hexokinase and 5 $U \cdot ml^{-1}$ glucose-6-phosphate dehydrogenase. The reaction was initiated by the addition of 3–5 ul of diluted (1:10) heart homogenate. The increase in absorbance was measured for 3 min at 340 nm before (AK activity) and after addition of 10 mM creatine phosphate (CK activity + AK activity). In the second run, 50 uM P1,P5-Di(adenosine-5')pentaphosphate was added to the buffer to inhibit AK. The absorbance was measured for 3 min at 340 nm before (to verify AK inhibition) and after addition of 10 mM creatine phosphate (CK activity).

The activities of CK and AK in the forward direction (ATP-consumption) were measured in a total volume of 1ml respiration buffer containing the following: 5 mM PEP, 2 mM ATP, 30 mM NADH, 5 U/ml PK, 2.5 U/ml lactate dehydrogenase. First, the absorbance was measured for 3 min at 340 nm with buffer to check whether the response is stable (to record the absorbance shift). Then 10 ul of diluted (1:10) heart homogenate was added (non-specific ATPase activity). Finally, the decrease in absorbance was measured after addition of 20 mM creatine (CK activity) or 1 mM AMP (AK activity).

The activities of CK and AK in both directions were measured with conditions similar to those used in respiration measurements and were calculated using the extinction coefficient for NADH/NADPH ($\epsilon 340 = 6.220$ $mM^{-1}$ $cm^{-1}$).

All enzyme activity measurements were performed in triplicate (in quadruplicate for CO) and the results averaged. Data was analysed using IOCBIO Kinetics [83].

## Determination of CK isoforms

The rat and mouse homogenates were also used to determine the CK isoform distribution as previously described [84]. The CK isoforms were separated by native agarose gel electrophoresis on a 1% agarose gel of approximately 1 mm thickness. The gel was transferred to a ceramic plate, which in turn was placed upon a cooling pad maintained at 15°C by a thermostat (Julabo F12-ED, JULABO Labortechnik GmbH). A small strip of filter paper was used to mark the row of loading spots about 1/3 from the anode. Drops of 1 µl homogenate (corresponding to tissue extract from 50 µg heart wet weight) were put on each loading spot with rat and mouse samples put one after another. Electrophoresis buffer was added to the anodic and cathodic

chamber, and wicks consisting of 4 layers of filter paper were used to connect the gel to the buffer in the two chambers. The gel was run at 250 V for 120 min.

The gel was transferred to the imaging chamber of an ImageQuant 400 (GE Healthcare Life Sciences). A filter paper the size of the gel was soaked in 5 ml visualization buffer and put on top of the gel. As the solution from the filter paper entered the gel, CK in the gel catalyzed the first of a chain of reactions: phosphocreatine + ADP + H+ → creatine + ATP. ATP from this reaction was used by hexokinase in the reaction: ATP + glucose → ADP + glucose-6-phosphate. Finally, glucose-6-phosphate was used by glucose-6-phosphate-dehydrogenase in the reaction: glucose-6-phosphate + NADP+ → 6-phospho-D-glucono-1,5-lactone + $H^+$ + NADPH. The increase in NADPH from the assay coupled to CK was followed by transillumination with UV-light and image capture with a SYBR Green filter. After ~20 min, the filter paper was carefully removed to image the NADPH signal in the gel. The gel pictures were analyzed using ImageJ software. Each lane was marked with a rectangle, and the area of each intensity profile peak corresponding to Mi-CK, MM-CK, MB-CK and BB-CK was noted. The relative intensity of each isoform was calculated.

The gel electrophoresis buffer consisted of (in mM): Tris 60, Tricine 60, EGTA 1, dithiotreitol 1, Triton X-100 0.1%, pH 8.6.

The visualization buffer consisted of (in mM): N-Acetyl-L-cysteine 120, phosphocreatine 120, glucose 70, MgAcetate 50, MES 22, ADP 9, β-Nicotinamide adenine dinucleotide phosphate 9, P1P5-Di(adenosine-5′) pentaphosphate pentasodium salt 0.2, pH 7.4. Immediately before use, hexokinase and glucose-6-phosphate dehydrogenase were both added to the visualization buffer to a final concentration of 5 IU/ml.

### Determination of Cyt aa$_3$ content

The Cyt aa$_3$ content in isolated rat cardiomyocytes was determined using a spectroscopic method independent of myoglobin contamination. This assay relies on the selective reduction of mitochondrial cytochromes by the action of sodium cyanide [85]. The rat cardiomyocytes were solubilized with 5% TritonX-100 in 0.1M potassium phosphate buffer (pH 7.5). The first graph (oxidized cytochromes) was obtained by scanning from 500 to 650 nm using an Evolution 600 spectrophotometer (Thermo Fisher Scientific Inc.) equipped with a Peltier water cooled cell changer (SPE 8 W; Thermo Fisher Scientific Inc.) to maintain temperature at 25˚C. The second graph (reduced cytochromes) was obtained the same way after reduction with 2 mM sodium cyanide in the presence of 15 mM ascorbic acid. The differential absorbance (reduced versus oxidized cytochromes) at 605 and 630 nm was used for quantification of respiratory chain Cyt aa$_3$ content (cytochrome c oxidase), using the extinction coefficient $\varepsilon 605 = 18.6$ mM$^{-1}$ cm$^{-1}$ [86].

### Normalization

For better comparison of our results to the findings of earlier studies, the respiration rates from permeabilized cardiomyocytes were normalized to protein content, Cyt aa$_3$ content (rat cardiomyocytes only), and CS activity. Enzyme activities in homogenates were normalized to wet weight and CS activity. The protein content in cardiomyocytes was measured spectrophotometrically in a BioSpec-nano (Shimadzu Scientific Instruments Inc., Columbia, MD) as previously described [21].

### Statistics

The values are shown as mean ± standard error of the mean (SEM). Statistical analysis of most data was performed using unpaired Student's t-test using R. However, the difference between

$V_{O2\_CK}$ and $V_{O2\_max}$, and $V_{O2\_AK}$ and $V_{O2\_max}$ was assessed using a paired t-test. Furthermore, for the comparison of $V_{O2}$ rates and $V_{ADP}$ rates (Tables 4 and 5, respectively), the effect of species and substrate was analysed by a mixed type ANOVA. $p < 0.05$ was considered statistically significant.

The raw data are given in the S2 Table.

## Supporting information

**S1 Table. Maximal respiration rate, $V_{O2\_max}$, and maximal ADP-phosphorylation rate, $V_{ADP\_max}$, normalized to the protein content of the cell suspensions, and $V_{O2\_max}$, normalized to the cytochrome aa$_3$ content.** The maximal respiration rate, $V_{O2\_max}$, was recorded in the presence of either GM or GMPS, and 2 mM ADP (see representative recording in Fig 3A). The rate was normalized to the protein content and, for rat cardiomyocytes only, the cytochrome aa$_3$ content. For statistical purposes, only the results from CK recordings are given. The corresponding maximal ADP-phosphorylation rate, $V_{ADP\_max}$, was calculated assuming P/O$_2$ ratios of 6 and 4 for GM and GMPS, respectively (see main text). Values from 8 mice and 7 rats are shown as mean ± SEM. * denotes $p < 0.05$, ** $p < 0.01$, *** $p < 0.001$, **** $p < 0.0001$, significant effect of substrate, species, or interaction between substrates and species.
(DOCX)

**S2 Table. Raw data from the experiments.**
(ODS)

**S1 Raw images.**
(PDF)

## Author Contributions

**Conceptualization:** Rikke Birkedal.

**Formal analysis:** Jelena Branovets, Marko Vendelin, Rikke Birkedal.

**Funding acquisition:** Marko Vendelin, Rikke Birkedal.

**Investigation:** Jelena Branovets, Kärol Soodla, Rikke Birkedal.

**Methodology:** Jelena Branovets, Rikke Birkedal.

**Supervision:** Rikke Birkedal.

**Visualization:** Jelena Branovets, Marko Vendelin.

**Writing – original draft:** Jelena Branovets, Rikke Birkedal.

**Writing – review & editing:** Jelena Branovets, Marko Vendelin, Rikke Birkedal.

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
