## [Decision Letter · Decision Letter 0]

12 Sep 2023

PONE-D-23-17486Rat and mouse cardiomyocytes show subtle differences in creatine kinase expression and compartmentalizationPLOS ONE

Dear Dr. Birkedal,

Thank you for submitting your manuscript to PLOS ONE. After careful consideration, we feel that it has merit but does not fully meet PLOS ONE’s publication criteria as it currently stands. Therefore, we invite you to submit a revised version of the manuscript that addresses the points raised during the review process.

We look forward to receiving your revised manuscript.

Kind regards,

Luis Eduardo M Quintas, Ph.D.

Academic Editor

PLOS ONE

Reviewers' comments:

Reviewer's Responses to Questions

**Comments to the Author**

1. Is the manuscript technically sound, and do the data support the conclusions?

Reviewer #1: Yes

2. Has the statistical analysis been performed appropriately and rigorously? 

Reviewer #1: Yes

3. Have the authors made all data underlying the findings in their manuscript fully available?

Reviewer #1: Yes

4. Is the manuscript presented in an intelligible fashion and written in standard English?

Reviewer #1: Yes

5. Review Comments to the Author

Reviewer #1: Review for PLOS ONE 09/06/2023

Title: “Rat and mouse cardiomyocytes show subtle differences in creatine kinase expression and

compartmentalization”

The manuscript described the subtle differences in CK expression and compartmentalization between rat and mouse cardiomyocytes. The methodologies are clear and sound, with convincing data.

There are some concerns about the manuscript in its present form:

1. Since the whole heart homogenates and isolated cardiomyocytes might have different characterizations, please explain the possible difference(s) and how the two could be integrated together.

2. There are extensive similar (or same) descriptions amongst the main text, figure legends, and method section. It would be better to integrate them to make the manuscript clear and concise.

3. Since the extensive usage of abbreviations, please move the Table of Abbreviations to the front, and define the abbreviations when they appear the first time. Also, please add Cyt aa3 to the list.

4. Line 59-60: Please add the reference and characterization of AGAT and GAMY mouse strains.

5. Line 142-145: Did the enzymatic activities normalize to the “heart” wet weight?

6. Fig 2B: The BB-CK bands are hardly recognized. It might be helpful to enhance the image.

7. Line 246: Please define P/O2 value.

8. Line 750: “Drops of 1ul homogenate”, does the homogenate normalized to the same protein concentration?

9. Please add a paragraph to explain the normalization method(s) that make the measurements are comparable between mouse and rat model.

6. PLOS authors have the option to publish the peer review history of their article (what does this mean?). If published, this will include your full peer review and any attached files.

Reviewer #1: No

---

## [Author Response · Author response to Decision Letter 0]

24 Oct 2023

The response to reviewers is attached as a file.

---

## [Editor Report · Decision Letter 1]

7 Nov 2023

Rat and mouse cardiomyocytes show subtle differences in creatine kinase expression and compartmentalization

PONE-D-23-17486R1

Dear Dr. Birkedal,

We’re pleased to inform you that your manuscript has been judged scientifically suitable for publication and will be formally accepted for publication once it meets all outstanding technical requirements.

Kind regards,

Luis Eduardo M Quintas, Ph.D.

Academic Editor

PLOS ONE
---

## [Editor Report · Acceptance letter]

14 Nov 2023

PONE-D-23-17486R1 

Rat and mouse cardiomyocytes show subtle differences in creatine kinase expression and compartmentalization 

Dear Dr. Birkedal:

I'm pleased to inform you that your manuscript has been deemed suitable for publication in PLOS ONE. Congratulations! Your manuscript is now with our production department. 

Kind regards, 

on behalf of

Dr. Luis Eduardo M Quintas 

Academic Editor

PLOS ONE